# Towards optogenetic vision restoration with high resolution

**Ulisse Ferrari** [ID]◉, **Stéphane Deny** [ID]¤◉, **Abhishek Sengupta** [ID]◉, **Romain Caplette** [ID], **Francesco Trapani** [ID], **José-Alain Sahel** [ID], **Deniz Dalkara, Serge Picaud** [ID]‡, **Jens Duebel**‡, **Olivier Marre**‡*

Sorbonne Université, INSERM, CNRS, Institut de la Vision, 17 rue Moreau, F-75012 Paris, France

◉ These authors contributed equally to this work.
¤ Current address: Department of Applied Physics, Stanford University, Stanford, California, United States of America
‡ SP and JD also contributed equally to this work.
* olivier.marre@inserm.fr

**Data Availability Statement:** The data are available here: https://zenodo.org/record/3820402#.Xrq6jBMzbOQ. The doi is: 10.5281/zenodo.3820402.

## Abstract

In many cases of inherited retinal degenerations, ganglion cells are spared despite photoreceptor cell death, making it possible to stimulate them to restore visual function. Several studies have shown that it is possible to express an optogenetic protein in ganglion cells and make them light sensitive, a promising strategy to restore vision. However the spatial resolution of optogenetically-reactivated retinas has rarely been measured, especially in the primate. Since the optogenetic protein is also expressed in axons, it is unclear if these neurons will only be sensitive to the stimulation of a small region covering their somas and dendrites, or if they will also respond to any stimulation overlapping with their axon, dramatically impairing spatial resolution. Here we recorded responses of mouse and macaque retinas to random checkerboard patterns following an in vivo optogenetic therapy. We show that optogenetically activated ganglion cells are each sensitive to a small region of visual space. A simple model based on this small receptive field predicted accurately their responses to complex stimuli. From this model, we simulated how the entire population of light sensitive ganglion cells would respond to letters of different sizes. We then estimated the maximal acuity expected by a patient, assuming it could make an optimal use of the information delivered by this reactivated retina. The obtained acuity is above the limit of legal blindness. Our model also makes interesting predictions on how acuity might vary upon changing the therapeutic strategy, assuming an optimal use of the information present in the retinal activity. Optogenetic therapy could thus potentially lead to high resolution vision, under conditions that our model helps to determinine.

## Author summary

In many cases of blindness, ganglion cells, the retinal output, remain functional. A promising strategy to restore vision is to express optogenetic proteins in ganglion cells. However, it is not clear what is the resolution of this new light sensor. A major concern is that

**Funding:** This work was supported by ANR TRAJECTORY, by the European Union's Horizon 488 2020 research and innovation programme under grant agreement No. 785907 489 (Human Brain Project SGA2), NIH grant U01NS090501, a grant from AVIESAN-UNADEV to OM, a Union's Horizon 488 2020 research and innovation programme grant No. 639888 to DD, a Foundation Fighting Blindness grant to S.P, D.D. and 491 J.A. S., ERC Starting Grant (OPTOGENRET, Grant No 309776) to JD and ERC 492 Starting Grant REGENETHER to D.D., RHU Usher (RHU LIGHT4DEAF [ANR-15-RHU-0001]) to S.P and D. D, IHU FOReSIGHT (ANR-18-IAHU-01) to S.P., and by the French State program 493 Investissements d'Avenir managed by the Agence Nationale de la Recherche 494 [LIFESENSES: ANR-10-LABX-65]. S.D. was supported by a PhD fellowship \DIM 495 cerveau et pensee" from the region Ile-de-France. This work was completed with the 496 support of the Programme Investissements d'Avenir IHU FOReSIGHT 497 (ANR-18-IAHU-01). The funders had no role in study design, data collection and analysis, decision to publish, or preparation of the manuscript.

**Competing interests:** The authors have declared that no competing interests exist.

axons might become light sensitive, and a focal stimulation would activate a very broad area of the retina, dramatically impairing spatial resolution. Here we show that this is not the case. Ganglion cells are activated only by stimulations close to their soma. Using a combination of data analysis and modeling based on mouse and non-human primate retina recordings, we show that the acuity expected with this therapy could be above the level of legal blindness.

## Introduction

The majority of inherited retinal degenerations are due to photoreceptor cell death. In many cases ganglion cells are spared making it possible to stimulate them to restore visual function in blind patients. Thanks to electrical implants it has been shown that stimulating retinal cells can evoke percepts [1–3]. A major challenge is to restore vision with a high resolution. A high acuity is necessary to solve complex visual tasks, e.g. face recognition, or navigating in complex environments (but see [4]). Several studies have suggested that expressing light sensitive proteins in ganglion cells could be an efficient way to restore vision [5–10], by stimulating these newly light-sensitive cells with patterned light to evoke visual perception. This is a promising alternative, but it is unclear what acuity can be expected with this strategy.

Direct measurements of acuity using behavioural tests have been performed on mice [9]. Measuring the acuity with behavioral experiments on non-human primates is challenging because of the lack of blind primate models. The light stimulation necessary to activate the transfected cells will also activate the photoreceptors in normal animals, and the effect of photoreceptor versus optogenetic activation would not be easily separated. To make progress towards predicting the acuity achievable by patients, an alternative is to quantify the spatial resolution of a retina treated with optogenetic therapy. For this purpose, one needs to measure the receptive field of ganglion cells, i.e. the region of visual space that will evoke a response. Large receptive fields should correspond to a poor resolution, while small ones suggest a high resolution, that could lead to a high acuity. Previous studies using optogenetic proteins expressed in ganglion cells mostly measured responses to full field flashes [5–9]. Some used fine-grained stimulation patterns that are necessary to measure receptive fields [11, 12], but only in the rodent retina, not in the primate retina. A few studies where light sensitivity was restored at the bipolar or photoreceptor stage used spots of increasing sizes to determine size selectivity [13, 14]. Overall, systematic estimation of receptive fields has not been performed previously in primate ganglion cells that are made light-sensitive. Since the optogenetic protein is also expressed in axons, it is unclear if these neurons will only be sensitive to the stimulation of a small region covering their somas and dendrites, or if they will also respond to a stimulation of their axon far away from the soma, dramatically impairing spatial resolution.

Here we developed an approach mixing experiments, data analysis and modeling to estimate the spatial resolution of retinas reactivated with optogenetics. In both mouse and macaque, optogenetic proteins were expressed in retinal ganglion cells following an intravitreal adeno-associated virus (AAV) injection *in vivo*. We then recorded the activity of populations of ganglion cells with multi-electrode recordings. We measured the receptive field of each cell, and built a quantitative model that recapitulates precisely how reactivated ganglion cells transform a stimulus into spike trains. This model acurately predicts the ganglion cell responses to complex stimulation patterns. We then used this model to estimate the spatial resolution of the reactivated retina. This resolution gives a prediction of the best acuity that could be reached by treated patients. We found that this acuity is above the limit of legal blindness. This result

suggests that therapy based on optogenetic reactivation is a promising avenue to restore high-resolution vision in blind patients. Our model also makes interesting predictions about different possible strategies to increase acuity of treated patients. Our approach is a first step towards predicting the acuity of a blind patient treated with optogenetic therapy.

## Results

### Optogenetically activated ganglion cells have localized receptive fields

We targeted retinal ganglion cells (RGCs) of blind rd1 mice (4/5 weeks old) with an AAV2 encoding ReAChR-mCitrine (a variant of Channel Rhodospin with red-shifted sensitivity) under a pan-neuronal hSyn promoter via intravitreal injections. Details of the gene delivery and optogenetic protein expression have been detailed elsewhere [7, 10]. Retinas were harvested after 4 weeks for multi-electrode array (MEA) recordings.

To estimate the size of the region whose stimulation can activate a ganglion cell (i.e. its receptive field), we displayed a random checkerboard stimulus (see Methods). We estimated the Spike Triggered Average (STA) by averaging over the frames that evoked a spike (see Methods) [15]. Many cells showed a well-defined receptive field (Fig 1A and 1D). Under the hypothesis of Gaussian shape, the average diameter of receptive fields was 99±7 $\mu$m (SEM, n = 30; individual uncertainty of 3 microns; Fig 1G and Methods), slightly smaller than what is usually measured in normal retinas [16]. We measured the temporal course of STAs (Fig 1B and 1E) with a checkerboard displayed at 40Hz. This frequency sets the limit of our temporal resolution in the estimation of the peak-response latency. We found the peak to be at the second time-bin, meaning that the latency is between 25 and 50ms. However, the temporal trace is above zero for the first time-bin, at about one third of the peak. This means that the onset of the response starts between 0 and 25ms, consistent with a direct light-activation of ganglion cells. See S1 Fig for more example cells.

To characterize the processing performed by the reactivated retinas, we built a classical Linear-Non-linear (LN) model [17] to predict the responses of ganglion cells to the checkerboard stimulus. The LN model is composed of a linear filter followed by a non-linearity to predict the firing rate of the cell. The linear filter was the average STA measured with the checkerboard. The non-linearity was learned on the checkerboard data using a classical maximum-likelihood estimation (see Methods).

We then tested the model on a repeated sequence of the checkerboard stimulus that was not used to learn the model. We restricted our analysis to ganglion cells that were strongly modulated by the stimulus (reliability score above 0.5, see Methods). On these cells, the LN model predicted very well cell responses to the checkerboard stimulus (Fig 1C and 1F for two examples). For all these cells, the prediction performance was high (mean Pearson correlation 0.69 ± 0.08, n = 24), close to the limit set by the reliability of the response (Fig 1H and Methods).

We then tested if the same approach could be applied in the non-human primate retina. We used data from a macaque retina where ganglion cells were transfected with the optogenetic protein CatCh (a variant of Channel-rhodopsin) [10]. Here AAV2 vectors were produced with a promoter driving the expression of the CatCh protein in ganglion cells. Macaques were injected intravitreally with AAV particles. Retinas were harvested 12 weeks after injection for MEA recordings. Expression of the optogenetic protein was restricted to the foveal ring [10], where most ganglion cells are midget cells [18].

To suppress the endogeneous photoreceptor response, retinae were light exposed for several hours, and LAP4 was added to the bath to block ON responses from photoreceptors (see [7] and Methods). We only found responses to light stimulation in the foveal ring, where the

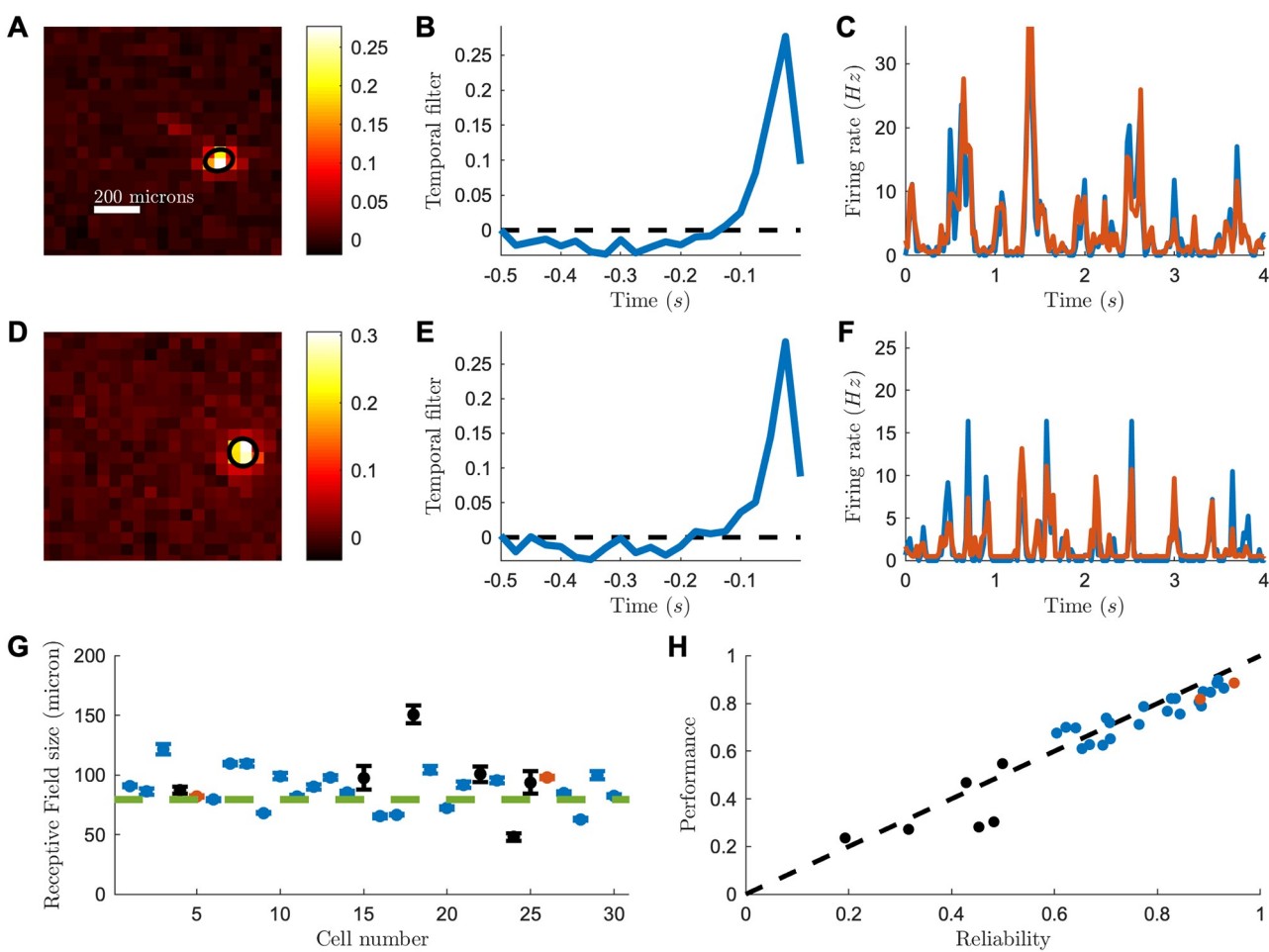

**Fig 1. Receptive field and LN model for a ReaChR reactivated mouse retina.** (A) Spatial receptive field of one example RGC, computed from the spike-triggered average (STA) to a checkerboard stimulus. Black ellipse is the 1-std contour of a gaussian fit to the STA. Scale bar: 200 microns. (B) Temporal receptive field computed from the STA of one RGC (see Methods). (C) Average firing rate (blue) from a cell in response to a repeated sequence of the checkerboard stimulus and predicted firing rate (red) from a LN model fitted to the cell response (r = 0.82). (D-F) as before, but for a different cell. (G) Receptive field sizes for the recorded cells. Red: example cells of the upper panels. Blue: highly reliable cells. Green line: weighted mean. (H) LN model performance plotted against cell reliability.

protein was expressed [10]. We performed a similar experiment as for the mouse, displaying a random checkerboard and measuring receptive fields.

Although the activity modulation by visual stimuli was much weaker than for the mouse (spontaneous activity was high, probably an effect of the culture or the bleaching), in some ganglion cells we could still find a STA with a well-defined receptive field (Fig 2A and 2D) with an average diameter of 82 ± 1 $\mu$m (SEM, n = 25; individual uncertainty of 7 microns; Fig 2G and Methods). We found in most of the cells that the peak-response was in the first time-bin, i.e. a latency smaller than 33.3ms and sometimes in the second time bin, i.e. a latency smaller than 66.6ms. Using the same strategy as in the mouse retina, a LN model could predict well the responses to repeated sequence of a checkerboard stimulus on cells with a clear modulation of the firing rate (Fig 2C and 2F, mean Pearson correlation 0.79 ± 0.1, n = 15), and was close to the performance expected given the reliability of the response (Fig 2H). See S2 Fig for more example cells.

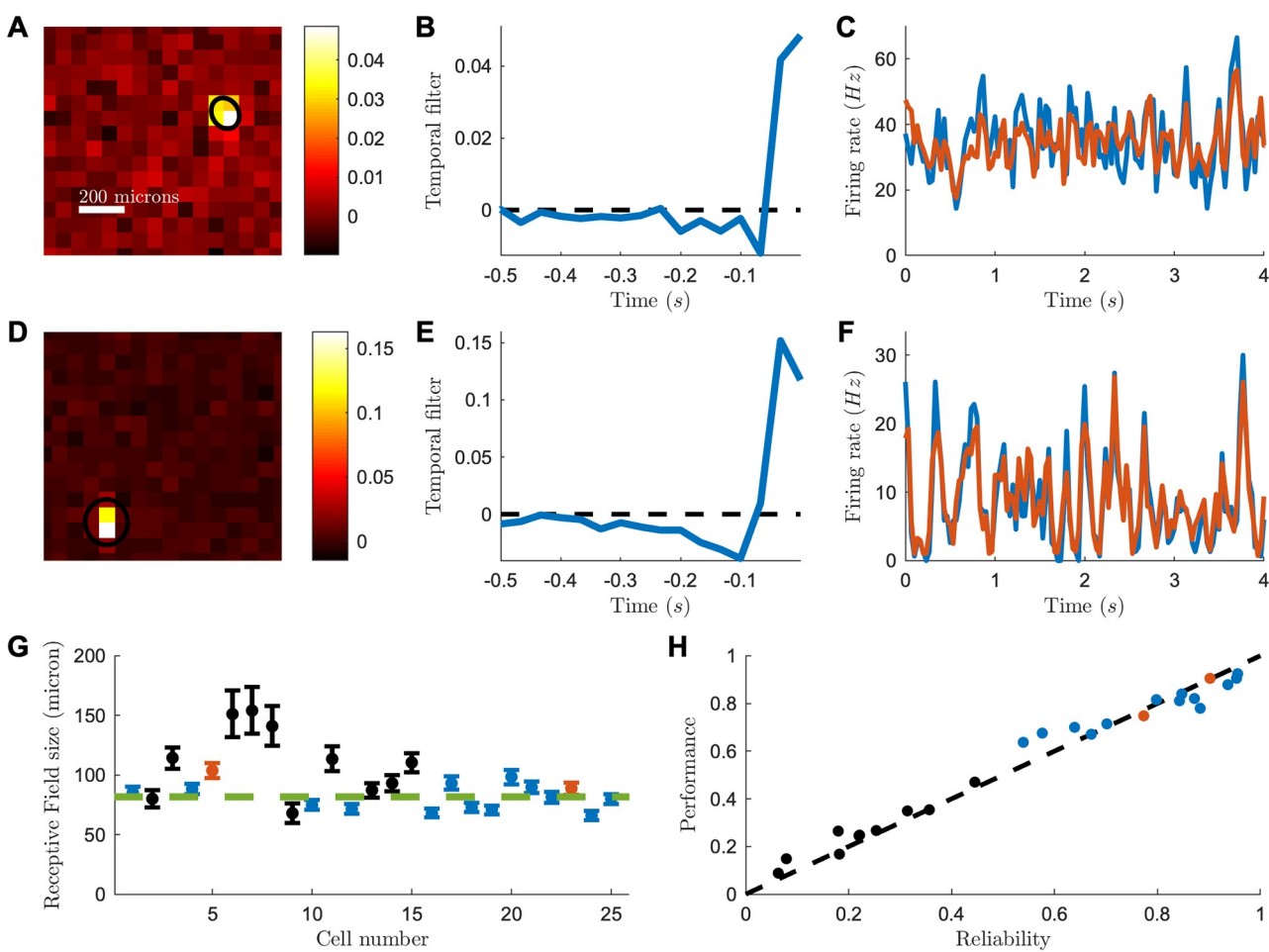

**Fig 2. Receptive field and LN model for a CatCh reactivated macaque retina.** (A) Spatial receptive field of one example RGC, computed from the spike-triggered average (STA) to a checkerboard stimulus. Black ellipse is the 1-std contour of a Gaussian fit to the STA. Scale bar: 200 microns. (B) Temporal receptive field computed from the STA of one RGC (see Methods). (C) Average firing rate (blue) from a cell in response to a repeated sequence of the checkerboard stimulus and predicted firing rate (red) from a LN model fitted to the cell response (r = 0.77). (D-F) as before, but for a different cell. (G) Receptive field sizes for the recorded cells. Red: example cells of the upper panels. Blue: highly reliable cells. Green line: weighted mean. (H) LN model performance plotted against cell reliability.

## Acuity estimation of the reactivated retina

Intuitively, the receptive field size is related to the spatial resolution of the retina. This spatial resolution is connected with the acuity that a blind patient treated with this optogenetic strategy could achieve. However, one would like to make this connection more quantitative. Since our model gives a precise account of how ganglion cells respond to visual stimuli, we can construct a full model of how the complete retinal population respond to a stimulus, and simulate the spike trains that the brain will receive from retinal ganglion cells. Thanks to this model, we can estimate the smallest letter size that can still be discriminated by an observer that would have access to these simulated spike trains. This smallest discriminable letter size is a good proxy of the best acuity reachable thanks to this therapeutic strategy. To construct a full model of the reactivated retina we assumed that ganglion cells were placed on a square grid, with a density equal to the density of transfected cells in the experiment (note that using a more irregular grid did not change significantly the results). In a previous study, we found that around

40% of ganglion cells were transfected in the macaque foveal ring (measured from confocal imaging in [10]), and the density of ganglion cells in the macaque fovea has been estimated to 51108 cells/mm$^2$ [19]. Each neuron was simulated with an identical LN model, with the parameters (STA, non-linearity) chosen to be equal to the average parameters found in the experimental data (see Methods). Each neuron in our model was thus identical up to a translation of its receptive field.

We then used our model to simulate the spiking response of the reactivated retina to an acuity test (Fig 3A). We chose a classical acuity test used in ophthalmology, the random E test, where the letter 'E' is presented in 4 possible directions. The test consisted in presenting randomly a letter to the retina in silico for 1 second, animated by a random jitter mimicking eye movements (see Methods), which corresponded to repeating the flashes multiple times. We then predicted which letter was presented from the spike trains using a Bayesian decoder (see Methods). By performing a Bayesian inversion of the model, it was possible to estimate which letter was presented. Using this decoder is equivalent to the assumption that the brain made the best use of the information contained in the spike trains received from the retina (a classical "ideal observer" hypothesis).

We simulated the activity of ganglion cells responding to letters E with different orientations and performed Bayesian decoding at different times following the beginning of the stimulation. As expected the success rate increased with time (Fig 3B). This is because the decoder accumulated evidence over time to better discriminate letters, and got better at finding which letter was presented when it had access to longer responses. Performance increased with the size of the letter to be decoded (Fig 3C). To be realistic and consistent with real in-situation acuity tests, we then defined the acuity score as the smallest letter size for which the success rate was above 80% within a time exposure of 1 second. We found that the smallest discriminable letter size was 110 microns. Assuming that discriminating a letter of 25 $\mu$m gives a 20/20 acuity, these 110 microns correspond to an acuity of 20/72. This is above the threshold of legal blindness (20/200).

## Discussion

### Localized receptive fields

A possible limitation of vision restoration with optogenetic reactivation of ganglion cells is that the whole axon of the neuron could become light sensitive, thus creating an unnaturally elongated receptive field, impairing resolution and therefore acuity. We have measured the receptive fields of retinal ganglion cells of mouse and primate retinas that were made light sensitive using an optogenetic protein.

[11, 12] and [15] have measured receptive fields in the mouse retina where ganglion cells are transfected by an optogenetic protein. However, this was not performed in the primate retina. Similar to humans, macaques have a fovea, i.e. a region of high resolution, while mice do not. To estimate the acuity that could possibly be reached by a human patient treated with this strategy, measuring receptive fields in the primate retina is therefore necessary. We found that many cells had a small receptive field. This demonstrates that the reactivated ganglion cells are mostly sensitive to the stimulation of their soma and dendritic field (and possibly the axon initial segment), but not to the stimulation of their axon, at least not the part of the axon that is the most distant from the soma: otherwise we would have measured large and elongated receptive fields encompassing the axonal image, and extending over several hundreds of microns. We cannot exclude that there would be an axonal response at much greater light intensities. Yet we demonstrated the existence of an intensity range at which cells can be stimulated close to their soma, but are not stimulated by their distant axon.

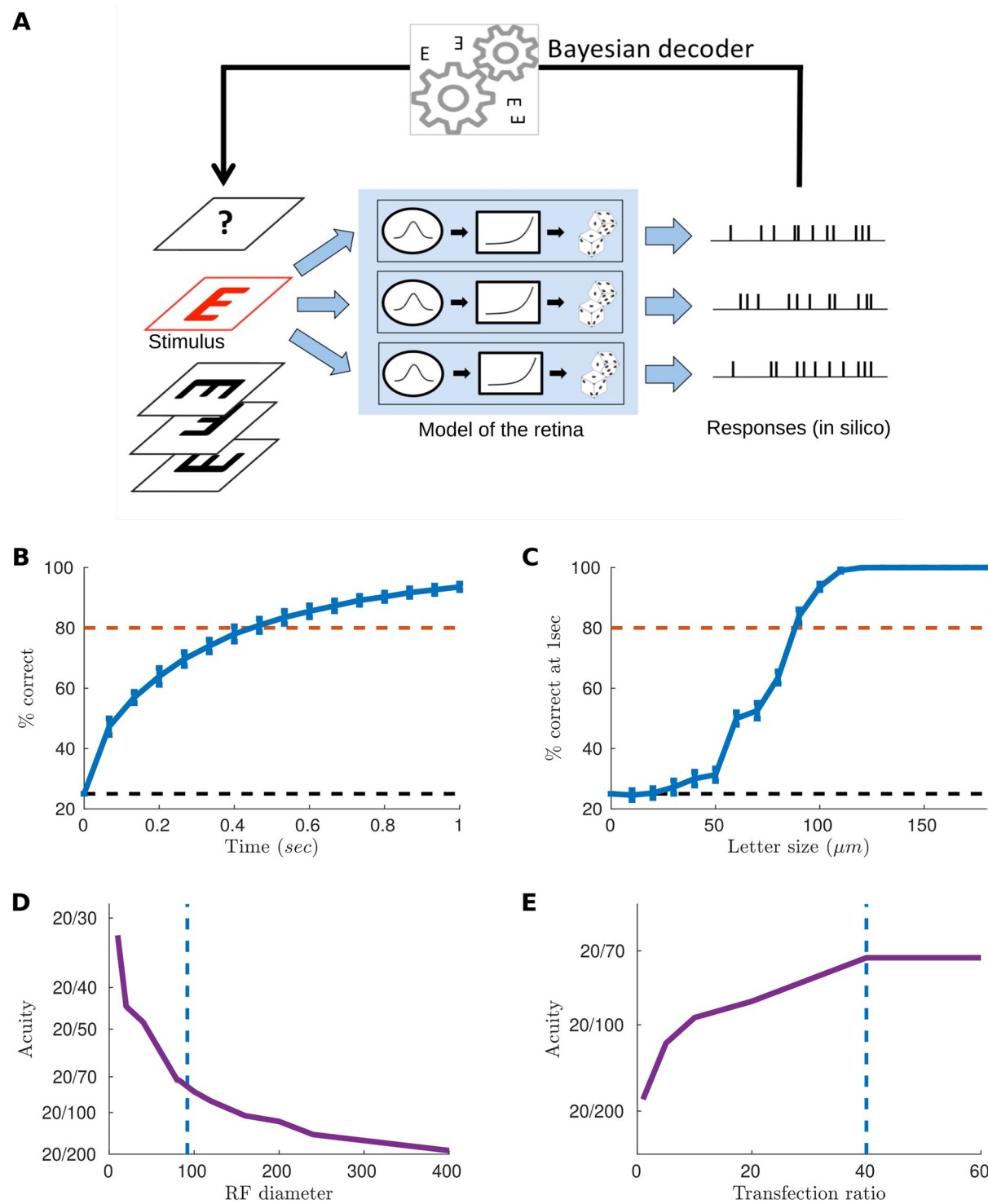

**Fig 3. Simulation of an acuity test *in silico* and estimation of acuity.** (A) We showed the letter E in one of 4 possible orientations and flashed the letter at a new random position every 33 ms to mimic fixational eye movements. "Time" refers to repetitions of these flashes. We then simulated the entire macaque retinal output as a collection of ganglion cells with receptive fields regularly spanning the visual field, all parameters being fitted to the experiments. To decode which letter was presented from these spike trains, we adopted a maximum likelihood strategy: we assumed perfect knowledge of the model and tested for which stimulus the observed spiking response was the most likely. We then chose the letter with the highest log-likelihood as the prediction of our model. (B) Average success rate over time, for a letter of 100 microns. (C) Percentage of letters decoded correctly after a 1s exposure, against the letter size. The dashed line is the 80% accuracy limit that we used in our definition of acuity (see Results). (D) Predicted acuity as a function of the receptive field size. (E) Predicted acuity as a function of the density of reactivated ganglion cells.

Several works aiming at restoring vision using retinal prostheses showed that electrical stimulation by epiretinal implants, one of the leading solutions to restore blindness, could activate distant ganglion cells, hundreds of microns away, through the stimulation of their en passant axons [20, 21]. As a result, this means that a neuron can be activated by very distant electrodes, and that its "electrical receptive field" is very broad, covering several hundreds of microns. It has been suggested that this distant activation of ganglion cells is the main limiting factor for the acuity reachable with retinal implants [21, 22]. Here we show that a similar issue does not seem to occur with our optogenetic strategy.

## Limits of the ideal observer analysis

To quantify the spatial resolution of the reactivated retina, we built a model of the information transmitted by reactivated ganglion cells to the brain. A similar approach using optimal Bayesian decoding has been applied to retinal models before [23], but was not applied to data where ganglion cells were made light sensitive with optogenetics for a therapeutic purpose. The assumption of an ideal observer implies that the patient has learned how to make an optimal use of the information transmitted by this reactivated retina.

The purpose of this approach is to fill the gap between the sensitivity of single ganglion cells, and the perceptual sensitivity that could be obtained with this therapeutic strategy. Previous studies [12] characterized single neuron sensitivity in the mouse as a proxy for perceptual sensitivity. However, many studies (e.g. [24]), have shown that care should be taken when comparing the sensitivity of single neurons, and the perceptual sensitivity. They demonstrated that, in general, there is no straightforward link between one and the other (see also [25]). These studies also established how modeling can be used to connect neural responses and perception, by building a model of the entire neural population responding to the stimulus, and quantifying how much information can be read out of this population. This is the approach we have followed here. Note that there is a specific advantage in doing this modeling of the population in the case of ganglion cells reactivated by optogenetics because we could find a simple model that predicts well ganglion cell responses to stimulation patterns, something which would be much more challenging with a normal retina.

[11] also tried a direct decoding of neural population activity. However, it is important to note that only part of the full population was recorded: if one wants to estimate the sensitivity of the full population of retinal ganglion cells, which is what the brain can access, modeling is necessary to avoid missing information provided by cells that were not recorded during the experiment. On one hand, direct decoding of the recorded population, like in [11], has the drawback of missing the activity of the cells that are not recorded. This can be an important issue when trying to estimate the resolution of the full reactivated retina. On the other hand, direct decoding allows skipping the use of a model. If the model is wrong, it could introduce errors in the estimate of the resolution. We have tested the performance of our model and shown that it predicts ganglion cell responses well, making the resolution derived from our model plausible. Another important difference between our approach and the one from [11] is the ensemble of possible stimuli when decoding the retinal responses. In the case of [11], the possible stimuli are the different letters presented. In our case, following [23], we also had the position of the letter as an additional variable, unknown to the decoder. At each time step, the position could change, mimicking the impact of eye movement (an approach also used in [23]). As a result, the possible stimuli are thus all the possible letters in all the possible locations. This is a very large ensemble, and to infer which letter was presented, one needs to also average the posterior over the possible positions in our Bayesian decoding. This is a more challenging problem, but closer to the problem that the brain needs to solve when only accessing

the ganglion cell activity, not knowing where the letter is a priori because of fixational eye movements.

Our model was realistic and properly fitted to the data, and allowed quantitative predictions of the best acuity we can expect in a treated patient. Our model predicted that a patient should be able to discriminate letters of size corresponding to a visual acuity of 20/72. The main assumption in this approach is that the brain can make the best use of the information transmitted by the retina. Even for late blind patients who have already experienced visual stimulation, the stimulation received from the reactivated retina will have a novel structure that needs to be learned. For example, former OFF ganglion cells now respond to light onset, and need to be processed like ON ganglion cells. Learning to use this novel retinal code will require a reorganization of visual cortices, where ON and OFF subregions have a distinct topographic arrangement [26, 27].

Several studies showed that the brain might not be able to make the best use of all the signals coming from the different cell types [28, 29]. However, recent works (see, for example, [30]) have shown that neurons might integrate the output of several cell types together, as early as in the LGN. It is therefore unclear if one should only take into account the output of a single cell type, or if the brain can pool the results from several cell types together. A possible consequence is that patients will only use the information coming from ON cells. In our model, this means that only half of the reactivated cells will send usable information, which is equivalent to divide the density of reactivated cells by two. Our prediction is that the impact of this division will be moderate, and acuity will still be above the legal level of blindness (Fig 3E).

Beyond the division between ON and OFF cells, our strategy based on AAV will transfect all the cell types and make them all have similar responses to light. It is unclear if the brain will make an optimal use of the activity coming from different cell types. In our data, the transfected ganglion cells were densely packed, and even when GFP was expressed together with the optogenetic protein, we could not isolate the dendritic tree of single cells. For this reason we cannot tell if different cell types have been transfected, but it is likely that the AAV used here will transfect all types of ganglion cells. However, in the transfected region, the foveal ring, most ganglion cells are midget cells [18]. Most of our transfected cells are thus likely to be midget cells. Restricting our decoder to using only the midget type would thus not lead to an important loss in resolution.

The reactivated ganglion cells form a ring around the photoreceptors of the fovea [10], and this novel geometry also needs to be learned by the brain. Many studies have shown that a reorganization of the adult visual cortex is possible following a lesion in the retina [31, 32]. Additionally, a promising strategy is to pre-process the visual input before sending a stimulation pattern to the ganglion cells. This pre-processing can be optimized to help the brain make the best use of the information transmitted by the retina. Both pre-processing of the visual input and brain plasticity could help to achieve a perceptual performance close to the optimal spatial resolution estimated here, but only direct tests on patients will determine how close to optimal they can be.

## Predicting efficacy of future treatments

An advantage of our modeling strategy is that it allows varying the different parameters to predict how they will influence the acuity. Here we have tested the impact of two relevant parameters in this model: the density of cells and the size of the receptive field. These two parameters will change depending on the therapeutic strategy adopted.

The size of the receptive field could change depending on how protein expression is engineered. [33] and [34] have shown novel engineered opsins can restrict their expression in the

somas. This could reduce the receptive field diameter and dramatically enhance acuity. Thanks to our modeling approach, we could estimate how acuity should change as a function of the RF diameter (Fig 3D). Very high acuity levels, above 10/20, could be reached if the receptive field diameter gets close to soma size (i.e. around 10 microns). However, if this comes at the cost of a lower expression level, resulting for example in a lower transfection ratio, this performance could be mitigated.

The density of cells will vary if the ratio of transfected cells is varied. This could happen when the AAV dose is changed, if a different capsid, promoter or optogenetic protein is used. We have estimated how the acuity will change when the transfection ratio is changed (Fig 3E). Surprisingly, the predicted acuity is only marginally affected by the transfection ratio, if all the other parameters of the model are kept equal. This is due to the high density of ganglion cells around the fovea: even with a 10% ratio a lot of cells will be activated, and this will be enough to transmit information. However, below this ratio performance decreased significantly. Our study thus emphasizes that several factors play a role in the expected acuity. Our approach integrates these different factors into a coherent framework and has thus the potential to help refining future optogenetic strategies by predicting the expected acuity in various conditions.

However, it is also worth noting that changing these factors with a different therapeutic strategy may also impact other aspects of this therapeutic approach. For example, changing the capsid could be more immunogenic [10], a risk associated with gene therapy [35]. Other factors need therefore to be taken into account when trying to find the optimal combination of capsid, promoter and protein to increase resolution [10].

In conclusion, our study shows that the optogenetically activated ganglion cells have a small receptive field. This localized processing enables them to deliver a high resolution picture of the visual scene to the brain. Our results suggest that optogenetic therapy based on in vivo AAV injection targeting ganglion cells should give an acuity above the limit of legal blindness. This would be a significant improvement compared to current strategies based on retinal implants, where acuity has remained below that level so far ([1, 2]; but see [3] for an alternative approach). Our approach can also be used to predict the impact of refined optogenetic strategies on the predicted acuity, and should be useful for further improvements of this therapeutical strategy.

## Material and methods

Data and code for reproducing the results are available at XXX. Unless stated otherwise, all error bars in figures and text are standard deviations over the samples. SEM stands for standard error of the mean and SD for standard deviation.

### Ethics statement

According to French regulation, the experimental protocol has been submitted to project authorization by the French Ministry of Research and Education (EU directive 2010/63), which includes the review by the local ethical committee (CETEA, n˚44), before the project begins, and got approval number: APAFIS#3180-201512150952286v4.

NHPs are observed daily and are in positive contact with experimenters by regular manipulation and distribution of treats to limit the possible impact of stress on physiological factors. The Local Animal Welfare Offices are in charge of improving animal welfare in housing and use of animals. These groups provide, for example, advices for implementation of standard operating procedures, humane endpoints, refining pain or distress management and environmental enrichment.

The animals are housed in pairs or in groups in individual cages in accordance with the revised Schedule A guidelines of ETS 123, adjacent cages have tactile and visual contact devices, groups are maintained in a visual, sound and tactile community. Animals are housed and accustomed to the environment at least four weeks before the operation. The animals have water ad libitum, access to the soil with litter, and the food distributed twice a day includes expanded croquettes and fresh, dried fruit. In addition to socialization, enrichment is ensured by food search devices and toys.

For surgery and longitudinal examinations, induction of anesthesia: (i) Ketamine mixture 10mg / kg and xylazine 0.5mg / kg IM after fasting the day before the experiment. (ii) Maintenance of the anesthesia: Propofol IV 1ml / kg / h for a better pupil dilation with oxybuprocaine (Cebesine) for local anesthesia of the eye. The animal is intubated for surgeries. Antibiotic / anti-inflammatory: framycetin / dexamethasone (Fradexam) local is applied after surgery.

Anesthetic overdose: pentobarbital sodium (180mg / kg) administered in IV after chemical restraint with ketamine (10mg / kg) / xylazine (0.5mg / kg), followed or not by intracardiac perfusion of physiological saline followed by paraformaldehyde at 4% according to the needs of the post-mortem analyzes.

### AAV production and injection

Details of the gene delivery and optogenetic protein expression in mice has been detailed elsewhere [7, 10]. Briefly, we targeted retinal ganglion cells (RGCs) of blind rd1 mice (4-5 weeks old) with an AAV2 encoding ReaChR-mCitrine under a pan-neuronal hSyn promoter via intravitreal injections. Four weeks post-injection, we observed strong retinal expression of mCitrine in rd1 mice as revealed by in vivo fundus imaging. For macaques, we targeted retinal ganglion cells (RGCs) with an AAV2 encoding a human codon optimized CatCh under a strong, RGC-specific promoter [10]. Retinas were harvested three months after injection of the virus in the adult macaque retina.

### Multielectrode array recordings

Recordings were made using a multielectrode array (MEA) comprised of 252 extracellular electrodes spaced at 100 $\mu$m on a square grid (Multi-Channel Systems, Reutlingen, Germany). Once a piece of retina had been isolated, it was placed ganglion cell side down onto the array. A perforated dialysis membrane was used to hold the retina in place on the array. The array was superfused with Ames solution (3 ml/minute, gassed with 95% $O_2$-5% $CO_2$) and maintained at 34˚C. Raw RGC activity was amplified and sampled at 20kHz. Resulting data was stored and filtered with a 200 Hz high pass filter for offline analysis. The recordings were sorted using custom spike sorting software developed specifically for these arrays [36, 37]. For macaque retina recordings, LAP4 was added to the bath to block ON responses coming from photoreceptors. This strategy has been validated previously [7].

### Visual stimulation and receptive field estimation

The stimulus was displayed using a Digital Mirror Device (supplier: *ViALUX*) and focused on the photoreceptor plane using microscope optics [38].

The receptive field (RF) of a retinal ganglion cell (RGC) is the particular region of the visual field in which a stimulus will trigger the firing of the cell. Here we characterized the spatial and temporal components of the RFs by estimating the spike-triggered average (STA) from a white noise checkerboard stimulus [17]. The stimulus was a flickering black-and-white checkerboard where the intensity of each checker was drawn at random from a binary distribution at every stimulus frame. The size of the checks was 67$\mu m$ for macaque and 50$\mu m$ for mouse, and frames

were updated at respectively $30Hz$ and $40Hz$. These coarse check sizes were used to maximize the Signal to Noise Ratio of the ganglion cell responses, but limited the precision of the receptive field estimation. The light source was an epi-fluoresence lamp with a white spectrum ranging from 380 to 780nm and a total light intensity of $10^{16}$ photons.cm$^{-2}$.s$^{-1}$. Computing the STA consists in selecting and averaging the frames in a 200ms time window preceding each spike, to form a 3 dimensional description of the receptive field (2 dimensions are space, 1 dimension is time). The spatial RF is defined as the temporal slice of the STA that contains the maximal value of the whole STA. The temporal RF is defined as the temporal evolution of the check of the STA with the maximal average value. To estimate the diameter of the receptive field we fitted a two-dimensional Gaussian to the measured spatial receptive field. In order to estimate the error of the RF size of each cell (due to the coarse check size mentioned above), we performed a boopstrap approach. We first generated many artificial noisy STAs by adding Gaussian noise (with zero mean and variance equal to that of the STA far from the cell center) to the fitted (and denoised) two-dimensional Gaussian. Then, by re-fitting this Gaussian on each noise-corrupted STA we estimated the standard deviation of the RF size. This strategy allowed us to quantify the error of individual RF sizes obtaining 3 and 7 microns for mouse and monkey, less than the spread of the sizes across the populations (Figs 1 and 2), which dominates the SEM estimates. As a consequence, even if our individual statistical errors were under-estimated due to the Gaussian hypothesis, the SEM estimation for the mean would not be much affected. Note, in fact, that these quantifications depend on the Gaussian assumption on the RF shape, which constraints the fit. This approach is standard in the retinal literature and consistent with the usual shape of RGC RF [17, 39].

Across all spike-sorted cells, we selected the cells that had a visible receptive field: 25/171 cells for the macaque, and 30/63 for the mouse, passed this test. The small ratio in the macaque is due to the fact that many recorded neurons were outside of the foveal ring, and therefore not transfected efficiently by the AAV [10]. In one of the macaque recordings where GFP was co-expressed with the optogenetic protein, the region transfected efficiently corresponds to around 35 electrodes (over 252) [10]. We then asked if cell responses were robust across repetitions of the same visual stimulus. To test for this, we divided stimulus repetitions in two halves and computed the mean response in time (PSTH) for each of them. Cells that showed a Pearson correlation between the two PSTHs larger than 0.5 passed the test: 15/25 for the macaque and 24/30 for the mouse.

## Linear Non-linear model

We fitted the responses of the ganglion cells with a Linear-Non-linear model [17]. In this model the stimulus is first convolved with the receptive field of the cell. Then the result goes through a parametrized *softplus* non-linearity ($\alpha \log(1 + \exp(\beta(x + \theta)))$) [38] to predict the firing rate over time. The non-linearity relates the amount of light in the receptive field to the firing rate of the cell, and was fitted by log-likelihood maximization.

To test the model, we repeated 50 times another sequence of the checkerboard stimulus, and we estimated the cells' firing rate by binning the response at 40Hz (mouse) and 30Hz (macaque). We then computed the firing rate predicted by our model and compared it to the experimental firing rate of the cell averaged across the 50 trials. To quantify the ability of our model to predict the cell response, we calculated the Pearson correlation (r) between the predicted ($f_{pred}$) and experimental ($f_{exp}$) firing rate. To measure the response reliability of a neuron, we split the repetitions in two halves and computed a PSTH on each half, and we then computed the Pearson correlation coefficient between these two PSTHs.

In the population model, the spatio-temporal RF shape was the same for all the neurons, only the center position of the cells differed. To estimate this averaged RF, we first computed the position of each recorded cell, then averaged the STAs of the re-centered cells. Then we decomposed this average RF into a spatial and a temporal component. The first was a two-dimensional, symmetric, Gaussian, whereas the latter was the RF temporal modulation at the center. For the non-linearity, we parametrized it as a soft-plus function with three parameters that we fitted with maximum log-likelihood from the response of all the cells to unrepeated checkerboard data.

## Acuity test in silico

To simulate the test, we picked one out of the 4 possible orientations and flashed the letter E at a new random position every 67 ms (corresponding to the decay time constant of the temporal RF). The purpose of this random renewal of the position was to mimic the fixational eye movements of the patient, which would displace the letter over the retinal surface (see also [23]). The letter was white on a black background. We assumed here that a patient would wear light amplifying goggles that would provide the light intensity necessary to activate the optogenetic protein.

We then simulated the entire retinal output as a collection of ganglion cells with receptive fields regularly spanning the visual field. We randomly picked a subset of cells that were the ones supposed to express the optogenetic proteins—the other ones were supposed to send no information about the stimulus and therefore removed: even if they have a spontaneous activity, it will be letter-independent and therefore it will affect minimally our ideal Bayesian observer. Each time a stimulus was presented, it was convolved with the receptive field of each cell and the result went through the non-linear function to predict the firing rate for each cell. We assumed that ganglion cells emitted spikes according to a Poisson process, as in a classical Linear-Non-linear Poisson model, previously used in the retina [40].

We then decoded which letter was presented from these spike trains. For this we adopted a maximum *a posteriori* strategy (with flat prior): we assumed perfect knowledge of the model and tested for which stimulus the observed spiking response was the most likely. First we computed the firing rates of the whole population of cells in response to every possible position and every letter f(cell, position, letter). Then, assuming a Poisson distribution of firing rates, we calculated the log-likelihood of the firing rates observed given any letter [41]:

$$\log p(f_{obs}|letter) = \sum_{position,cell} f_{obs}(cell) \log f(cell, position, letter)$$
$$-\Delta t \sum_{position,cell} f(cell, position, letter) + c \tag{1}$$

where $f_{obs}$ are the firing rates simulated for all cells in response to the letter presented, $\Delta t$ is the time of presentation of the letter in a given position (60ms), and $c$ is a constant.

We then chose the letter with the highest log-likelihood as the prediction of our model.

The decoding was performed at each time step. Over the time course of the presentation, the decoding could benefit from evidence accumulated at previous time steps. The decoding performance thus got better and better over time. We repeated this test 500 times with random letters and averaged the performance over time of our decoder. The performance was defined as the percentage of letters correctly guessed. We defined the acuity as the smallest letter for which the performance was larger than 80% after a time exposure of 1 second. To estimate the standard deviation in our results, we repeated the whole procedure 30 times.

The human eye is larger than the macaque eye and this could affect acuity for a receptive field of the same size: according to [42], one degree of visual angle corresponds to 275 microns in the human fovea, and 223 in the macaque fovea. However, the proper comparison should take into account both this factor and the relative size of midget cells in human and macaque retinas, by comparing dendritic field size converted in degrees of visual angle. This comparison has been performed [42] and showed that the sizes were very similar.

## Supporting information

**S1 Fig. Receptive field and LN model for a ReaChR reactivated mouse retina. (A-D): 4 more example cells.** Left: Spatial and temporal receptive field. Center: experimental (blue) and predicted (red) firing rate in response to a repeated sequence of the checkerboard stimulus. Right: filtered stimulus plotted against the experimental firing rate for each time-bin of the repeated stimulus. Red: prediction of the firing rate using the non-linearity function. (TIFF)

**S2 Fig. Receptive field and LN model for a CatCh reactivated macaque retina. (A-D): 4 more example cells.** Left: Spatial and temporal receptive field. Center: experimental (blue) and predicted (red) firing rate in response to a repeated sequence of the checkerboard stimulus. Right: filtered stimulus plotted against the experimental firing rate for each time-bin of the repeated stimulus. Red: prediction of the firing rate using the non-linearity function. (TIFF)

## Author Contributions

**Conceptualization:** Ulisse Ferrari, Stéphane Deny, Abhishek Sengupta, José-Alain Sahel, Deniz Dalkara, Serge Picaud, Jens Duebel, Olivier Marre.

**Data curation:** Ulisse Ferrari, Stéphane Deny, Abhishek Sengupta, Romain Caplette, Francesco Trapani.

**Formal analysis:** Ulisse Ferrari, Stéphane Deny, Abhishek Sengupta.

**Funding acquisition:** José-Alain Sahel, Deniz Dalkara, Serge Picaud, Jens Duebel, Olivier Marre.

**Investigation:** Ulisse Ferrari, Stéphane Deny, Olivier Marre.

**Methodology:** Ulisse Ferrari, Stéphane Deny, Olivier Marre.

**Project administration:** José-Alain Sahel, Deniz Dalkara, Serge Picaud, Jens Duebel, Olivier Marre.

**Software:** Ulisse Ferrari, Stéphane Deny.

**Supervision:** Serge Picaud, Jens Duebel, Olivier Marre.

**Validation:** Ulisse Ferrari, Stéphane Deny, Olivier Marre.

**Visualization:** Ulisse Ferrari, Stéphane Deny, Olivier Marre.

**Writing – original draft:** Ulisse Ferrari, Stéphane Deny, Deniz Dalkara, Serge Picaud, Jens Duebel, Olivier Marre.

**Writing – review & editing:** Ulisse Ferrari, Stéphane Deny, Deniz Dalkara, Serge Picaud, Jens Duebel, Olivier Marre.

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
