## [Decision Letter · Decision Letter 0]

28 Aug 2019

Dear Dr Marre,

Thank you very much for submitting your manuscript 'Towards optogenetic vision restoration with high resolution' for review by PLOS Computational Biology. Your manuscript has been fully evaluated by the PLOS Computational Biology editorial team and in this case also by independent peer reviewers.

While all reviewers thought the paper covered an interesting and important topic, all reviewers had major concerns with the current manuscript and requested major revisions. The most concerning issue, however, is that one of the reviewers also challenged the novelty of this work with reference to previous papers that was not cited in the current manuscript.

Unfortunately, when taking together these expert reviews, we cannot accept this article for publication in the current form. If the authors are willing to conduct an extensive revision and address all the listed major concerns and - most importantly - if they rewrite the manuscript in a manner that clearly outline the novelty of this study over previous work, we will consider this manuscript for publication. We cannot, of course, promise publication at that time.

Sincerely,

Michiel van Wyk

Guest Editor

PLOS Computational Biology

Lyle Graham

Deputy Editor

PLOS Computational Biology

[LINK]

Reviewer's Responses to Questions

**Comments to the Authors:**

Reviewer #1: Review of Ferrari et al. (2019) "Towards optogenetic vision restoration with high resolution", PCOMPBIOL-D-19-00958.

This manuscript is an interesting and fairly straightforward study on the receptive field size of ganglion cells in mouse and macaque retina that have been given photosensitivity using a variant of channel rhodopsin supplied by an AAV virus. The receptive field size was measured using a checkerboard stimulus to produce a spatio-temporal Spike-Triggered Average from recordings of isolated retinas using an MEA. A model of the transfected ganglion cell array was created from the average ganglion cell STA responses using a LN model that generated spikes with stochastic timing, placed on a square grid with spacing set from the density of transfected cells. The visual acuity was then predicted from Bayesian tests of the model's responses evoked by visual stimuli randomly jittered to simulate eye movements.

The manuscript is relevant to recent attempts to produce high-acuity vision for the blind using optogenetics. It seems apprpriate for the journal because it is an excellent combination of electrophysiology and computational modeling. Overall it is well-written and the figures are adequate and appropriate. However, I have several major concerns about the methods and the resulting acuity derived from the model, and several minor suggestions about wording.

1. Receptive field size. As mentioned in the abstract and main text, looking at Figures 1 and 2, the reader sees no evidence of the axon within the optogenetic receptive field (RF). But the small RF diameter, modeled in Figure 3D, seems a different issue. The measurement of RF size with a 50 um check size in mouse and 67 um check size in macaque seems difficult to reconcile with the estimated size of ~93 um. With the assumption that the receptive fields were circular (and e.g. Gaussian), it might be reasonable to measure such an average RF diameter only 40% larger than the check size. But with a stationary grating, many RFs would have likely been misaligned with the borders of the checks so the STA average could not accurately reflect the actual diameter or shape. Without the circular Gaussian assumption, such a coarse check size seems inappropriate because it would not allow accurately defining RF shape and size. If the checkerboard stimulus could move (e.g. with some jitter, as in the model) one could imagine a higher resolution than the check size. But to ask a question about the shape of the RF, i.e. to what extent the ChR is in the axon and/or dendrites, the 67 um check size would seem too coarse to determine RF shape and size. Although the reader wonders about the significance of the exact RF size on acuity (see point 2 below), the measurement of RF size seems questionable. For the purpose of reporting the results on RF size, it would be more convincing to show several of the RFs with finer scale plots, or even better, measured with finer check size.

2. The relation between receptive field size, cell spacing, and acuity is not well described. Large receptive fields, i.e. larger than the cell spacing, will obviously reduce effective acuity, but it is not clear how receptive fields much smaller than the cell spacing can improve acuity. Although Fig. 3D,E show acuity vs. RF diameter and acuity vs. transfection ratio (i.e. relative density), it is difficult for the reader to understand how acuity could be high (e.g. 20/40) with very small receptive fields (e.g. size of a soma, 10-20 um). Given a density of 51 cells/mm2 the average spacing in the real retina is ~140 um -- but with a transfection of 40%, the spacing of transfected cells is likely to be quite random and ~50% greater, ~220 um. With average spacing of 220 um it is difficult to imagine how receptive field sizes of less than 100 um, e.g. 10 - 50 um, could give improved performance in discriminating letter rotations.

The reader needs some explanation of how a receptive field the size of a cell's soma, in an array of cells spaced on average 220 um apart, could give higher acuity than one would predict from the spacing alone. It would seem that the term "hyper-acuity" should be applied in this case where the receptive fields are much smaller than the average spacing. Generally hyper-acuity described in the literature is only relevant for discriminating certain stimuli -- but usually not for letter rotations. It would help the naive reader to add a short section on hyper-acuity with appropriate references.

The main conclusion, that optogenetic vision could support useful high-resolution vision, seems based mainly on the lack of ChR sensitivity in the axon and the average cell spacing. The measurement of RF size and its relation to acuity seems less convincing.

3. Cell type. Although the issue of interpretation of the signals from ON vs. OFF cells by cortical circuitry is discussed, there appear to be other related issues that would be a problem in assessing the performance of the model. Since the cell types of the recorded cells have not been reported, and are likely unknown, the reader wonders whether large and small cells, e.g. magno vs. parvo types, or other types typically recorded in macaque, would be transfected -- and in what proportions. Different cell types are known to have different responses: e.g. transient vs. sustained -- and these if averaged would likely reduce the Bayesian performance. The question this raises is the relevance of the model with the assumption that different cell types are transfected and stimulated, and may project to different cortical areas that may not be able to directly integrate an acuity discrimination with each other. Therefore, it would seem that the acuity estimated is only an upper limit, and that the actual acuity from a real transfection would likely be substantially lower.

The issue of cell type also is relevant to the receptive field size, as it seems possible that different cell types could have different transfection patterns that would affect their receptive field sizes and shapes.

4. Grid geometry. It would seem that a semi-random grid geometry, set with the average nearest-neighbor distance and regularity from the real transfection pattern, instead of a square grid as contained in the model, would be more appropriate -- as then the model would more closely reflect the pattern of spacing of the real cells and possible hyper-acuity.

5. l 80, "Details ... has been detailed", suggest "Details ... have been detailed"

6. l 113, "due to photoreceptor", suggest "from photoreceptors"

7. l 113, "We performed a similar experiment than for mouse", suggest "We performed a similar experiment as for the mouse".

8. l 116, "we could still find STA with a well-defined receptive field in some ganglion cells (Fig 2A,B) with an average diameter of 92.8+/- 10 um", suggest "in some ganglion cells we could still find a STA with a well-defined receptive field (Fig 2A,B) with an average diameter of 93 +/- 10 um".

9. l 133, "were placed on a squared grid", suggest "were placed on a square grid".

10. l 137, "51,108 cells/mm2", suggest "51.1 cells/mm2". Although a comma is often used as a decimal point, for this journal a dot "." seems more appropriate.

11. l 147, "Using this decoder is equivalent to assume that", suggest "Use of this decoder is equivalent to the assumption that".

12. l 308, "and compute the mean response", suggest "and computed the mean response".

13. l 158-189 "these 110 microns corresponds", suggest "these 110 microns correspond".

14. l 308 "and compute the mean response", suggest "and computed the mean response".

15. l 326, "only the center of the cells", suggest "only the center position of the cells".

16. l 330, suggest placing "Deny et al." inside parentheses: "(Deny et al., 2017)".

17. l 337 "is to mimic", suggest "was to mimic".

18. l 339, A white letter on a black background is generally not considered to be 100% contrast, as the overall level of illumination would depend on the fraction of the visual field that is covered with white. In real experiments, the background level must be measured and reported, as it sets the adaptation level in the visual system. A truly black background would dark-adapt the retina and therefore the white letter stimuli would generate an undetermined level of illumination, an undetermined level of adaptation, and undetermined contrast. This effect might not affect the results from the model -- but it should be described in the text so as not to confuse the reader.

Reviewer #2: Review is uploaded as an attachment

Reviewer #3: In their manuscript 'Towards optogenetic vision restoration with high resolution', the authors address an interesting topic. The analysis is currently quite limited. Especially, given the complexity and value of the primate study, the authors should present their data in a less qualitative way. The authors should be able to improve the value of their study significantly without the need for additional experiments.

The paper is to a great extent motivated by the fact that the whole cell becomes light-sensitive. However, this is hardly quantified in the manuscript. The variance of a Gaussian fit does not represent the possible influence of a light-sensitive axon appropriately. In figure 1A some sensitivity is visible at 11 o'clock. Is this a common observation? Is this in the direction of the axons? Is the SNR of the measurements sufficient to see axons? One could imagine e.g. an estimation based on the volume of the soma and axon, and the size of the stimulus grid.

The measured receptive field diameters are much larger than those of the somas'. How strong is the influence of the stimulus grid? If the primary dendrites play a significant role, a significant variation across cell-types might be observable. Are microscopic images of the cells available? In the primate, a significant number of cells should be midgets and parasol cells with distinct differences. Do all RGCs share a similar time course?

'...which correspond to ganglion cells where the expression level of the optogenetic protein was high enough.' Please provide quantification.

'The small ratio in the macaque is due to the fact that many recorded neurons were outside of the foveal ring, and therefore not transfected efficiently by the AAV...' Can this be quantified?

Please provide evidence that the untreated tissue showed no light responses.

'Assuming that discriminating a letter of 25 μm gives a 20/20 acuity'. Please provide citations to go from the macaque to the human.

'The letter was white on a black background (100% contrast)' A lower contrast might be a more realistic assumption.

'...the other ones were supposed to send no information about the stimulus and therefore removed' They should have a spontaneous background rate, possibly even a correlated firing pattern.

'...flashed the letter E at a new random position every 67 ms' Fig 3B should be labeled 'repetitions' or the model should use a continuous random walk as stimulus input.

Please extend your discussion of the fact that the model is close to the best-case scenario. Possibly, tone down the abstract as this study analyses three cells and extrapolates them to 50000/mm2.

How many separate recordings are part of this paper?

Is data used in other publications?

What data is shared?

Overall, please provide more quantifications and citations throughout the manuscript.

'Here we developed an novel computational approach...' The computational approach is not new.

**Have all data underlying the figures and results presented in the manuscript been provided?**

Reviewer #1: Yes

Reviewer #2: Yes

Reviewer #3: No:

PLOS authors have the option to publish the peer review history of their article (what does this mean?). If published, this will include your full peer review and any attached files.

Reviewer #1: No

Reviewer #2: No

Reviewer #3: No

---

## [Decision Letter · Decision Letter 1]

25 Feb 2020

Dear Dr. Marre,

Thank you very much for submitting your manuscript "Towards optogenetic vision restoration with high resolution" for consideration at PLOS Computational Biology. The reviewers appreciated the extensive revision you have conducted and agree that the manuscript has gained significantly. Nevertheless, they still suggest some importnat modifications to your manuscript. Based on the reviews, we are likely to accept this manuscript for publication, providing that you address these final concerns of the reviewers.

Sincerely,

Michiel van Wyk

Guest Editor

PLOS Computational Biology

Lyle Graham

Deputy Editor

PLOS Computational Biology

[LINK]

Reviewer's Responses to Questions

**Comments to the Authors:**

Reviewer #1: Review of Ferrari et al. (2019) "Towards optogenetic vision restoration with

high resolution", PCOMPBIOL-D-19-00958_R1.

The authors have responded well to most of the reviewers' comments, as

their revision has added additional information that clarifies many of the

problems that had been identified. I only have minor suggestions for revision.

1. line 18, "might vary upon changing therapeutic strategy", suggest

"might vary with changes in the therapeutic strategy", or

"might vary upon changing the therapeutic strategy", or

"might vary depending on the therapeutic strategy".

2. line 20, "under conditions that our model helps determining.", suggest

"under conditions that our model helps to determine."

3. lines 89-90, Again I would suggest that the method used could not determine

the receptive fields with the accuracy described. The problem as previously

explained is the shape of the receptive field assumed in fitting the diameter.

Suggest changing "98.7+/-7" to "99+/-7", or even "100+/-7".

Suggest adding a comment about the "individual uncertainty of 3 micron," to

explain how this accuracy was determined based on the 50 um check size, and

add a short section detailing the assumptions about the receptive field shape.

In Methods a section (lines 394-398) mentions the use of a fitted

two-dimensional Gaussian, but it would be helpful to explain why this would be

appropriate for the image of a soma and proximal dendrites labeled by the AAV

virus, and to explain the expected error by assuming a Gaussian shape..

Suggest change "of 3 micron" to "of 3 microns".

These revisions will not affect the manuscript's conclusion, but would be

less confusing to the naive reader who might wonder how such an accuracy

could be given with a check size of 50 um.

4. line 96, change "start" to "starts".

5. line 96, "in line with a direct light-activation", suggest

"consistent with a direct light-activation".

6. line 121, change "photoreceptor" to "photoreceptors".

7. line 127, "we could still find STA with", suggest

"we could still find a STA with".

8. line 128, as above, please explain how this accuracy in the diameter of the

receptive field was determined based on the 67 um check size, and add a short

section (maybe the same as for mouse, above) detailing the assumptions about

receptive field shape.

9. line 159, "which corresponds to repeat the flashes multiple times", suggest

"which corresponded to repeating the flashes multiple times" or

"which was equivalent to repeating the flashes multiple times."

10. line 185, "has not been performed", suggest, "was not performed".

11. line 188, "showed a small", suggest "had a small".

12. line 194, "larger light intensities", suggest "greater light intensities",

or "brighter light intensities".

13. line 195, "cannot be stimulated", suggest "are not stimulated".

14. line 208, "but was never", suggest "but was not".

15. line 209, "optogenetics in a therapeutic purpose", suggest

"optogenetics for a therapeutic purpose".

16. line 241, "one need to also", suggest "one needs also to".

17. line 271, "will transfected", suggest "will transfect".

18. line 385, "but limits the precision", suggest "but limited the precision".

19. line 395, "a Gaussian", suggest "Gaussian".

20. line 400, please explain "salient receptive field", or choose different

wording, as the naive reader is confused by this phrase.

21. line 440, "enable to reach the light intensity", suggest

"provide the light intensity".

22. line 446, " and therefore it will not", suggest "and therefore will not".

23. line 446, "therefore it will not affect our ideal Bayesian observer".

I suggest that, strictly speaking, this is incorrect, as the ideal observer

must take into account the variability in each cell by shifting ideal choices

slightly away from the letter-independent noise distribution. Suggest

"therefore it will minimally affect our ideal Bayesian observer" or similar wording.

24. lines 469-473, discussion of going from macaque to human: would be helpful

here to mention the relative size of the macaque and human eyes, and how this

would affect the comparison. The same size ganglion cel receptive field will

give higher acuity in a larger eye (i.e. with a greater post-nodal distance).

Reviewer #3: The author clarified many concerns, added additional data, quantifications, and analyses and thereby strengthened the paper. However, I would have hoped that the authors take a more careful approach and further tune down their claims.

I doubt that the area of additional sensitivity in figure 1A is caused by 'random fluctuation of noise'. It might very well not be related to the axon, but it looks like what one might expect the RF of an initial axon segment to look like. Similar hints might be visible in figure S1 B and D. However, it is not strictly necessary to follow up on this.

I hoped my comments on the model would result in more realistic versions in addition to the best-case scenario. Of course, the contrast in a prothese would be quite high, but possibly not permanently binary. While the current optimal Bayesian decoder is not affected by the rhythmic firing of unstimulated neurons, a decoder that uses the same likelihood, independently of whether a cell is stimulated or not, would be. This might better resemble the visual system of a patient prior to relearning.

The description of the new analysis to estimate the error in the measured RF sizes due to coarse checkers is not clearly formulated. It seems that the authors doubled the background noise and refitted the Gaussian. I don't think that this is a good test. One possibility might be to subsample and then refit Gaussians of known size to simulate the effects of alignment and stimulus size.

**Have all data underlying the figures and results presented in the manuscript been provided?**

Reviewer #1: Yes

Reviewer #3: None

PLOS authors have the option to publish the peer review history of their article (what does this mean?). If published, this will include your full peer review and any attached files.

Reviewer #1: No

Reviewer #3: No
---

## [Editor Report · Decision Letter 2]

7 Apr 2020

Dear Dr. Marre,

We are pleased to inform you that your manuscript 'Towards optogenetic vision restoration with high resolution' has been provisionally accepted for publication in PLOS Computational Biology.

Best regards,

Michiel van Wyk

Guest Editor

PLOS Computational Biology

Lyle Graham

Deputy Editor

PLOS Computational Biology

---

## [Editor Report · Acceptance letter]

8 Jul 2020

PCOMPBIOL-D-19-00958R2 

Towards optogenetic vision restoration with high resolution

Dear Dr Marre,

I am pleased to inform you that your manuscript has been formally accepted for publication in PLOS Computational Biology. Your manuscript is now with our production department and you will be notified of the publication date in due course.

With kind regards,

Laura Mallard
